# A Normalization Protocol Reduces Edge Effect in High-Throughput Analyses of Hydroxyurea Hypersensitivity in Fission Yeast

**DOI:** 10.3390/biomedicines11102829

**Published:** 2023-10-18

**Authors:** Ulysses Tsz-Fung Lam, Thi Thuy Trang Nguyen, Raechell Raechell, Jay Yang, Harry Singer, Ee Sin Chen

**Affiliations:** 1Department of Biochemistry, National University of Singapore, Singapore 117596, Singapore; uly.lam@nus.edu.sg (U.T.-F.L.); nguyentrang071291@gmail.com (T.T.T.N.); bchrae@nus.edu.sg (R.R.); 2Singer Instruments, Roadwater, Watchet TA23 0RE, UK; jay.tc.yang@hotmail.com (J.Y.); harry@singerinstruments.com (H.S.); 3NUS Center for Cancer Research, National University of Singapore, Singapore 117599, Singapore; 4NUS Synthetic Biology for Clinical & Technological Innovation (SynCTI), Life Science Institute, National University of Singapore, Singapore 117456, Singapore; 5National University Health System (NUHS), Singapore 119228, Singapore

**Keywords:** drug screening, yeast, edge effect, high-throughput screening, *Schizosaccharomyces pombe*, fission yeast, high-density array, hydroxyurea

## Abstract

Edge effect denotes better growth of microbial organisms situated at the edge of the solid agar media. Although the precise reason underlying edge effect is unresolved, it is generally attributed to greater nutrient availability with less competing neighbors at the edge. Nonetheless, edge effect constitutes an unavoidable confounding factor that results in misinterpretation of cell fitness, especially in high-throughput screening experiments widely employed for genome-wide investigation using microbial gene knockout or mutant libraries. Here, we visualize edge effect in high-throughput high-density pinning arrays and report a normalization approach based on colony growth rate to quantify drug (hydroxyurea)-hypersensitivity in fission yeast strains. This normalization procedure improved the accuracy of fitness measurement by compensating cell growth rate discrepancy at different locations on the plate and reducing false-positive and -negative frequencies. Our work thus provides a simple and coding-free solution for a struggling problem in robotics-based high-throughput screening experiments.

## 1. Introduction

In vivo high-throughput cell-based screening is a popular protocol for biomedical research to understand genetic interactions [1,2,3], protein–protein interactions [4,5], microbial pathological mechanisms [6,7] and screening for candidate chemicals or drug targets for translational applications [6,8,9,10,11]. High-throughput microbial arrays evaluate the cell permeability [12], cytotoxicity [13,14] and targeting efficacy [15,16] of novel or existing chemicals. These assays often employ qualitative measurement of cell viability or quantification of colony size changes on solid agar media incorporated with designated drugs [3]. 

*Schizosaccharomyces pombe* (*S. pombe*) is a powerful ascomycetes model organism possessing metazoan-like physiological processes, including RNA interference pathway [17,18], human-like mitochondrial inheritance and energy metabolism [19,20,21], equal nuclear division through cell fission [21,22], regional centromere and other conserved genomic features [21,23], as well as a high level of functionally homologous proteomic compositions [24]. The short duplication time [25], well-characterized growth conditions and availability of complete molecular, biological and genetics toolkits in combination with the comprehensive gene knockout libraries [26] have promoted fission yeast to be an ideal model for high-throughput screening [21,27,28].

*S. pombe* is a favorable model for chemogenomic studies of chemotherapeutic agents to uncover modes of action to facilitate further application to humans [11,29,30,31,32,33,34,35,36]. Assessment of cell growth is an easy and cost-effective approach to derive cellular response to the drugs. Digital applications have been developed for quantification of colony growth fitness, for example, FitSearch [37], ImageJ-linked CellProfiler [38], Java-based HT Colony Grid Analyzer [39] and Python-based Colonyzer [40], which are able to infer the growth outcomes into drug responsive phenotypes. Some of these programs have been modified to be used in combination with automated robotic platforms, such as the ROTOR^®^ HDA (Singer Instruments, Somerset, UK) platform, which is widely employed for microbial high-throughput analyses, especially for yeasts, including *S. pombe*. The ROTOR system utilizes the PhenoBooth^®^ scanner to image colonies, and microbial colony growth is measured by the PhenoSuite^®^ software (version 2.21.0304.1) based on different parameters such as colony volume and size [39,40,41,42,43,44,45,46,47,48].

Notwithstanding all the technological breakthroughs, high-throughput screening efforts that involve colony growth measurement are confounded by the edge effect phenomenon. Hitherto, the edge effect remains an unresolved issue. Presumably, low density of microbial colonies situated at the edge of the agar plate permits additional surface area for cells to expand from fewer competing neighbors that may arise from higher nutrient availability to enable better cell fitness, which in turn results in bigger colony size [11,49,50,51,52,53,54]. Temperature gradient and differences in evaporation rate at different locations of agar plate and subsequent changes in local humidity may further impact reproducibility of colony size data [53,54]. 

An approach to minimize edge effect involves randomizing the pinning position of the biological repeats within media plates [11,53]. However, doing so would multiply the number of plates required for the experiments and thus increase the complexity of the screening layout (Figure 1). Analysis programs such as the ROTOR^®^ HDA-coupled PhenoSuite^®^ software (version 2.21.0304.1) host a normalization plug-in script to reduce edge effect by adjusting size values of colonies situated at the plate boundary with reference to average colony size within the same row or same plate [11,53,55,56]. This approach, however, is prone to overcorrection, which, if occurs, compromises the accurate interpretation of the drug hypersensitivity of yeast strains pinned at the plate boundary [53] (refer below, Appendix A).

In the attempt to find a simple way to address the edge effect issue, we characterized the growth of mutant and wild-type (WT) strains at different positions on solid agar media plates to reveal the frequency of false positives and false negatives. Our observations suggest that a major contributor to false results is the edge effect, and we identified a normalization ratio that can reduce the occurrence of these false data, especially in a 384-well format.

## 2. Materials and Methods

### 2.1. Drug

Hydroxyurea (HU) purchased from Sigma-Aldrich (Burlington, MA, USA) and dissolved in water prior to use was performed according to manufacturers’ recommendations. HU was chosen for its water solubility, chemical stability and heat resistance. Previously published hypersensitivity response of multi-environmental factors responsive (MER) strains to HU [34] revealed by serial dilution spotting assay was used as the standard to compare with the HU hypersensitivity response of the MER strains on the high-density array.

### 2.2. Fission Yeast Techniques

Manipulation of fission yeast cells follows standard protocol [57]. All yeast cultures were incubated in yeast-extract adenine (YEA) media with 3% glucose (Sigma-Aldrich, Burlington, MA, USA), 0.5% yeast extract (BD Biosciences, San Jose, CA, USA), and 75 mg/L of adenine (Sigma-Aldrich, Burlington, MA, USA), adjusted to pH 5.5, without additional humidification. MER strains described in this paper (Appendix A) were modified from the Bioneer deletion mutant library (version 2.0; Daejeon, Republic of Korea) as previously described [58]. The overall workflow of the experiments conducted in this study is shown in Figure 1.

### 2.3. ROTOR Manipulation and Establishment of Normalization Table

WT, HU-hypersensitive *Δcds1* (positive control) and *Δdad2* cells were pinned from a 96-well liquid source plate to an agar source plate with 96 long-pin Repads (Singer Instruments, Somerset, UK) using Singer ROTOR HDA robotics (Singer Instruments, Somerset, UK) (Figure 2). After 3-day incubation at 30 °C, colonies were then transferred from the agar source plate onto drug plates with 0, 2 and 4 mM of HU with Singer ROTOR HDA robotics (Figure 2). Images were captured with a Phenobooth^®^ scanner (Singer Instruments, Somerset, UK) with reflective white lighting at power = 0.54 units, brightness = 0.03 units, gain = 0 unit, exposure = 20.2 ms, hue = 0.5 unit, saturation = 0.5 unit, white balance (blue) = 0.5 and white balance (red) = 0.5 (Figure 2). Acquired images were then processed with PhenoSuite^®^ (Singer Instruments, Somerset, UK) software (version 2.21.0304.1), with background subtraction threshold = 150 units and “large colonies” mode, to generate colony size value at each position of drug plates (Figure 2 and Figure 3).

### 2.4. Microbial Array Assay

MER yeast cells (Appendix A) were first pinned from a 96-well plate exponentially growing liquid culture (OD_600_~0.5) to a master source plate with 96 long-pin Repads (Singer Instruments, Somerset, UK) using Singer ROTOR HDA robotics. After 3-day incubation at 30 °C, colonies were then transferred in quadruplicates from the agar source plate with 96 long-pin Repads (Singer Instruments, Somerset, UK) onto drug plates with 0, 2 and 4 mM of hydroxyurea for screening using Singer ROTOR HDA robotics. Drug plates were then incubated at 30 °C and scanned every two hours manually with PhenoBooth^®^ (Singer Instruments, Somerset, UK). Images captured were then analyzed with PhenoSuite^®^ (Singer Instruments, Somerset, UK) software (version 2.21.0304.1) to generate the colony size value at each position of the drug plates (Figure 2 and Figure 3). Average colony size values of biological replicates of MER strains on drug plates were plotted against incubation time to deduce the colony growth rate for later comparison with the value of the WT strain (Figure 3). To analogize ROTOR screening with traditional spotting analysis, growth rates were separately determined at two linear regions of the growth curve (i.e., 27th–46th hours of incubation after pinning (early growth rate) and 27th to 71st hours of incubation after pinning (late growth rate)) (Figure 3). Mutants having a slower growth rate than the WT strain are deemed “sensitive” to HU, while those with a faster growth rate than WT are “resistant” to HU.

### 2.5. Mathematical Analyses

All mathematical calculations involved in growth curve plotting or edge effect normalization were done by Microsoft Excel modules within the Microsoft 365 package (version 16.77.1) (Redmond, WA, USA).

Parameters for statistical analyses of ROTOR results are calculated using the following formula [59], with sensitive strains considered as positive results:$$Accuracy=\frac{Number\;of\;true\;positive\;strains+Number\;of\;true\;negative\;strains}{Number\;of\;true\;positive\;strains+Number\;of\;true\;negative\;strains+Number\;of\;false\;positive\;strains}\times 100\%$$
$$Precision=\frac{Number\;of\;true\;positive\;strains}{Number\;of\;true\;positive\;strains+Number\;of\;false\;positive\;strains}\times 100\%$$
$$False\;Postive\;\%=\frac{Number\;of\;false\;positive\;strains}{Number\;of\;true\;positive\;strains+Number\;of\;false\;positive\;strains}\times 100\%$$
$$False\;Negative\;\%=\frac{Number\;of\;false\;negative\;strains}{Number\;of\;true\;negative+Number\;of\;false\;negative\;strains}\times 100\%$$

## 3. Results

### 3.1. Subsection

#### 3.1.1. Time of Documenting Improves Description of Growth Outcome

Growth of fission yeast cells in a microbial array is usually evaluated based on their colony sizes or colony growth rates [39,40,41,42,43,45,46,58]. The ROTOR HDA system (Singer Instruments, Somerset, UK) is a high-throughput colony transfer robotic platform commonly employed for genome-wide screening that employs the whole-genome gene-knockout strain libraries. In a preliminary experiment to prepare for large-scale screening using this robotic platform, we attempted to quantify colony growth using 91 MER mutants identified in previous screens of several drugs including the ribonucleotide reductase inhibitor hydroxyurea (HU) [34,35,58,60]. The spotting experiments were robustly reproduced in our laboratory, especially on HU, thus establishing a reliable basis of comparison for the accuracy of high-throughput pinning array analyses. Similar results performed on budding yeast were unavailable to us for such standardization. As such, our study only focused on fission yeast.

We started by pinning WT and MER mutants exhibiting different degree—high, medium and low—hypersensitivity towards HU [34] in 96- and 384-array formats on YEA-rich agar media incorporated with 0, 2 and 4 mM HU, using the ROTOR HDA robot. These strains were revived from cryopreservation and amplified before being inoculated into YEA media in a source microtiter plate for overnight growth into log-phase growing cultures (OD_600_~0.5). A liquid-to-agar plate transfer was performed the following day with the ROTOR platform. These plates were allowed to grow for a further day, to ensure that cells were healthily amplifiable. Finally, a third step of agar-to-agar transfer was performed onto YEA agar plates that were incorporated with HU or control plates without HU. Cells were allowed to grow for 5 days on the agar plates, and growth was documented every two hours after pinning (Figure 2).

The MER strains were initially identified via manual serial dilution spotting assays [34,58,60]. However, we came to realize that it was highly challenging to visually differentiate the hypersensitivity of the strains from colonies that were pinned onto the agar plates (Appendix A). This issue consequently resulted in a high proportion of false positives and false negatives that did not recapitulate published results (Appendix A). The problem persisted even when the imaging solution was employed to quantify the growth outcomes of the colonies via analyzing the images captured with PhenoBooth^®^ (Singer Instruments, Somerset, UK) and analyzed with PhenoSuite^®^ software (version 2.21.0304.1) (Singer Instruments, Somerset, UK) (Appendix A). During imaging, we were met with an additional difficulty in determining the best time point to document the colony size so as to reliably analyze the growth data. In the process, we discovered that measurement of the growth rate over a stretch of time, instead of determining the absolute size of colonies at particular time points, yielded a more accurate description of the drug responsiveness of the strains (Figure 3A,B). We observed further that strains that grew slower were less likely to be differentiated at earlier time intervals after the strains were pinned onto the plates and hence should be documented at the later time point. The contrary was true for faster-growing mutants when an earlier time point can already differentiate their growth discrepancies (Figure 3B).

#### 3.1.2. Growth Discrepancy of Colonies at Designated Positions on Agar Plates

Next, we ask whether colony growth can be affected by locations within the media plate and, if so, whether that can confound the quantification and interpretation of strain hypersensitivity. We chose three strains—WT, *Δcds1* and *Δdad2* strains—that showed no, high and moderate sensitivity to HU, respectively, based on our previously reported serial dilution spotting assay [34] (Figure 3C). We pinned an entire plate with the colonies of one strain using the ROTOR HDA platform, at two different densities of 96 and 384, on plates incorporated with 0, 2 and 4 mM of HU. We paid particular attention to assessing how colony growth may be affected with varying the distance from the edge of the agar plate at different plating densities [34,58]. The PhenoSuite^®^ program (version 2.21.0304.1) was employed to image the colonies and quantify the size of the colonies.

First, we assessed the 96-colony density plates. Visually, on untreated plates, we detected observable differences in the sizes of colonies between those pinned at boundary position relative to those further inward towards the center after 69 h of incubation. This was observed for all three strains tested (WT, *Δcds1* and *Δdad2*). The extent of such edge effects on colony size diminished with higher drug concentration in plates (Appendix A).

Interestingly, the edge effect on 96-density plates became insignificant when we compared growth using relative colony growth rate, which we defined as the change in colony size per incubation period (in hours) normalized to WT cells (Figure 4 and Appendix A). Using this measurement, we further tested 91 MER strains high-density pinning at 96-format and noted a high level of reproducibility of >93%, by comparing to our previously reported spotting results (Appendix A), including for strains spotted at the edge of the plate. These observations indicate that using growth rate as a means of comparison removed the edge effect observed at this pinning density. This is presumably due to the colonies being still quite well spaced apart to prevent nutrient accessibility from becoming limiting.

Next, we repeated the test by pinning the colonies in 384-format. Unlike the case of 96-format, we observed prominent edge effects with colonies growing much better at the edge of the plate with regards to both colony size as well as colony growth rate. The effect was especially apparent after 45 h of incubation, regardless of the drug concentration or yeast strains used. The increased growth of the colonies was observed for two rows from the edge of the plates (Figure 5 and Appendix A). We quantified the growth rate of all the colonies on the plate at the early and late growth phases (Appendix A). We noticed that the growth of the colonies pinned at the four corners of the plate (Figure 5A: dark blue A1, P1, A384 and P384), the outermost row (Figure 5A: Outer Edge, pink, B1-O1, A2-A383, P2-P383 and B384-O384) and the second-most outer row (Figure 5A, Inner Edge, orange, B2-O2, B2-B383, O2-O383 and B383-O-383) grew significantly differently from those pinned at the 3rd and 4th row and columns (Figure 5A, yellow and green respectively) or further inside. The growth rate of colonies inner from the third row/column onward approximated that of the colonies pinned at the center of the plate (Figure 5A, black). For most of the strains at both early and late phases of growth, the corner colonies grew a generalized 4-fold better than those at the inner locations on the plate, whereas those at the outer edge and inner edge exhibited 3- and 2-folds better growth (Figure 5B and Figure 6A), which allowed us to construct a normalization table to correct the edge effect by dividing the growth rate of the colonies at the affected locations.

The better growth of the colonies situated at the edge of the plate may be attributable to higher accessibility to nutrients relative to that in the middle of the plates, which have more competing neighbors in the 384-format. We obtain further support for this hypothesis by intentionally omitting spotting in the middle of the agar plate. Interestingly, doing so created a new “edge” at that inner locality as we detected enhanced growth of colonies surrounding the empty space even though these colonies were physically located at the center of agar plates (Appendix A).

Taken together, these results showed that the degree of impacts of edge effects on colony size is inversely proportional to pinning density. Consequently, the propensity for misinterpretation of drug sensitivity in such tests will be more severe at higher colony pinning density. Considering the observation of the difference in growth rate at the corner versus the inner and outer edges of the plate (Figure 5), it will become detrimentally confounding if the positive (known sensitive strains) and negative (usually WT) control strains—to which all strains on the plates will be normalized—were placed at these locations (Appendix A). Such aberrations will then aggravate false-positive and false-negative interpretations of the growth results. While false positives can still be rectified by secondary tests, such as using conventional spotting assays, false negative strains will be entirely excluded from the test as they would never have been identified as drug-responsive strains in the first place.

#### 3.1.3. Normalization for HU Screening with MER Strains

We thus consolidated a normalization table based on the analyses performed in the attempt to rectify edge effects for array-based drug hypersensitivity screening utilizing colony growth rate. This table was derived by quantifying differences in colony growth rate values of WT, at all coordinates within the agar plate in 384-format (Figure 6A). Similar levels of growth differences were observed for Δ*dad2* and Δ*cds1* mutants and at early versus late phases of growth, suggesting that the observed discrepancy may be independent of strains and days at which documentation was performed, and further suggesting that normalization can be performed observable from the growth difference of the WT cells.

We assessed whether the screening efficiency may be improved by normalizing with the derived values (Figure 6A). To this end, we pinned the MER strains [34,60,61] to quantify their growth on agar media plates containing 4 mM of HU. The hypersensitivity phenotypes obtained were cross-compared with previously reported serial dilution spotting [34]. Four technical repeats of each MER strain were pinned at four different loci on the same agar plate to obtain an average growth rate for comparison of HU sensitivity. The proportions of false-positive and false-negative results were calculated at early and late phases of growth compared to intermediate (day 3) and stationary growth (day 7) from serially diluted spotting.

Among the MER strains, we were only able to repeatedly observe 74.4% and 67.8% of the pinned MER strains that show HU sensitivity in the high-density 384-format array in two trials (Accuracy % in Figure 6B and Appendix A). Upon applying the normalization, there was no change to the reproducibility using early growth rate data, but there we observed a 3.5% improvement in the accuracy of hypersensitivity phenotypes assessment when employing late growth rate data. On the other hand, precision %, which is calculated using the formula
$$Precision=\frac{Number\;of\;true\;positive\;strains}{Number\;of\;true\;positive\;strains+Number\;of\;false\;positive\;strains}\times 100\%$$

Ref. [59], was marginally improved after applying normalization to the growth rate data and yielded 2.5% and 1.7% using early and late growth phase data. False-positive rate appeared to track that of accuracy %, showing 0.3% improvement comparing early growth rate with day 3 spotting data, and 1.7% (5-fold that of early growth phase data) when comparing early colony growth rate with day 3 spotting results (Figure 6B). Surprisingly, false negative % showed a much better improvement than was unexpected from the apparently small changes in the accuracy % and precision %: 4.4% and 9.8% improvement after the application of normalization to early and late growth rate data in comparison to serial dilution spotting data at day 3 and day 7, respectively (Figure 6B).

Taken together, our work yielded a tailored normalization table that can allow simple mathematical transformation to improve the efficiency of genome-wide screening of fission yeast using high-density array, particularly to reduce false negatives, which hitherto is difficult to quantify and hence modify.

## 4. Discussion

### 4.1. Challenges of Colony Size Normalization via Image Processing Procedures

High-throughput colony-based screening has been widely used in biomedical and industrial research [3,8,28,62,63,64]. Microbial colonies robotically pinned on agar plates are quantitatively monitored with a camera to capture cellular growth in terms of colony size and other derived values (e.g., relative colony growth rate, colony volume, colony opacity (OD_600_) [39,40,41,42,43,44,45,46]. Microbial assays based on colony size or its derivative are affected by changes in non-biological independent variables including humility during incubation and scanning [44,65], light artifacts during scanning [40,44,47], source plate-originated bias [44,51], pinning density [44,48,52] and most importantly edge effects [11,44,49,50,51,52]. The extent of edge effects is likely correlated with pinning density, which complicates the proper determination of colony characteristics (Figure 4 and Figure 5).

Software tools have been optimized to counteract the aforementioned issues via image processing and/or computational correction [39,40,42,44,47,51,52]. However, most of these programs adopt an assumption of “normal distribution of fitness effects”, in which ranges of colony sizes are assumed to follow a Gaussian distribution, and biological treatments including drug applications and genetic backgrounds are assumed to have little impact on fitness scores of genetic mutants [44,52]. The adoption of such overgeneralized assumptions may create bias, especially at the level of presetting of the normalization protocol for analyzing the microbial assay screening results by the software, leading to negligence of small differences in fitness scores and reduced screening resolution [44,66]. ROTOR-coupled PhenoSuite^®^ adopts a similar normalization protocol to determine and correct outlier values by comparing with the average colony sizes across the same row of colonies or the same agar plate [11,53,55,56].

Recently, some protocols were developed without assumption of the “normal distribution in fitness effects” such as the linear interpolation-based (LI) detector tool, yet even so, it has not completely addressed edge effect issues, as the values derived from colonies at the edge or corner, which were known to be suffered from the effect, were automatically removed from calculation instead [44]. Doing so will result in considerable wastage as a significant portion of the results has to be discarded (if considering our findings, the two outer rows and columns contain 144 colonies among a total of 384, which amounts to 37.5% of the total number of colonies that need to be removed). Furthermore, complicated data filtering procedures with LI detector require extensive prior knowledge in computation, which may not be user-friendly to biomedical scientists working in a wet lab setting. Other attempts were carried out to construct a normalization protocol based on certain statistical models, which can likely also cause adjustment of raw data value by >30% to incorporate further biases arising from such “over-correction” of high-throughput screening results [53].

Another big challenge, though subtle, is how to standardize the exact number of cells between different strains robotically pinned at different positions of agar plates. A potential solution to this issue was proposed by making use of a 2-step “punch-in” protocol, in which yeast cells from liquid media are first pinned at low pinning pressure onto a thicker source plate, and thereafter colonies grown are then transferred from the source agar to test plate at normal thickness using higher pinning pressure. Observations obtained from “punch-in”’ manipulation were reported to avoid edge effects in 384-format arising from changes in colony opacity (OD_600_) [46]. We tried this method, although it was unable to achieve elimination of the edge effect via the measurement of the colony growth rate. We surmised that this may be due, at least in part, to the variation in the growth when different strains were revived from cryopreservation, thus resulting in uneven amplification and in turn causing difficulty for colonies to be equally picked up by the pin. This motivates us to revive the cells in liquid media, transferring the revived strains through a solid media before finally pinning onto the test plates. Indeed, doing so has solved this issue in our hands.

### 4.2. Imaging Quantification Versus Serial Dilution Assay

In view of the lower reliability of existing high-throughput microbial array systems, conventional spotting remains a golden standard for yeast-based drug screening in recent decades [67,68,69,70,71]. Efforts have been made by scientists to optimize high-throughput microbial assay with reference to spotting-based viability assay. Pinning serially diluted fission yeast cells with robotics may be one of the possible alternatives to improve accuracy in terms of small-scale high-throughput assay. However, it will be extremely tedious if we consider manual dilution steps for an entire deletion library, which in the case for *S. pombe* may contain close to 4000 strains without the use of automated screening [26]. Conversely, the use of liquid-handling platforms can be employed to prepare such serially diluted source plates. However, these platforms are typically costly and will preclude access for most labs owing to funding limitations.

### 4.3. High-Density Array Analysis versus Plate Reader-Based Continuous Growth Measurement

Our analyses revealed that timing of growth documentation is critical for an accurate interpretation of growth phenotype and drug hypersensitivity responses of strains. One might then consider an alternative approach of growth measurement through continuous monitoring of optical density using a plate reader to avoid the need to determine the critical time for optimal sampling of colonies grown on agar media. Plate reader-based approaches are more technically straightforward, easy to use and fast. However, there are several important drawbacks when compared with the approach employing high-density pinning array. Firstly, the small circumference of the microtiter plate wells prevents thorough mixing of the cultures posing sedimentation of the microbial cells as an issue that confounds accurate measurement of optical density, and special technical modifications are often required to overcome this issue [72]. Even though normal plate readers are equipped with a shaker function, depending on the density of screening, very-high-speed shaking of the plates is often needed and may increase the chance of spillage and cross-contamination of cultures in adjacent wells. Plates with fewer but bigger wells are often preferred to minimize the intensity of shaking. This is however achieved at the expense of throughput. Hence, the high-density protocol reported herein combines the strengths in terms of ease of use, augmented throughput and accuracy of sampling results.

### 4.4. Advantages of Colony Size Correction with a Normalization Table

The normalization pipeline we have innovated in this work requires only simple normalization without the need for algorithm transformation. As such, no prior knowledge in programming nor robotics is required, which is useful for use by researchers who are unfamiliar with computational coding. It also does not require additional hardware or software modules, and hence is compatible with any known system of high-throughput microbial screening—manual or automated—to derive meaningful data that can approximate the accuracy of the serial dilution-spotting assay. Users are able to empirically tailor-make their normalization table at the start of the screening with novel chemicals. This can be achieved by performing a preliminary pinning test of WT and known sensitive strains onto the drug agar plates, to quantify colony growth pattern and overall colony growth rate ratio across the plate using the principles we have outlined herein. Users can then follow up with actual pinning test runs to ensure the magnitude of edge effect and the variation from the general recommendations proposed by our work, simply with the use of Microsoft Office Excel^®^ software (version 16.77.1). We believe this approach is also generically applicable to high-throughput assays using other prokaryotic and eukaryotic microorganisms as an answer to medical questions and/or industrial applications, which will be revealed in future empirical testing [2,63,64].

## Figures and Tables

**Figure 1 biomedicines-11-02829-f001:**
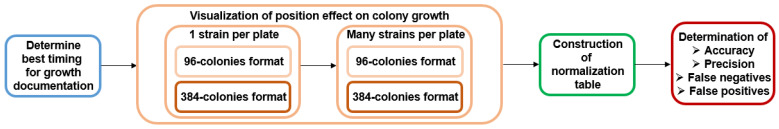
Flowchart showing an overview of experiments conducted in this work. Overall, there are four steps: (1) starting from the determination of the best timing and approach to document colony growth on the high-density array, (2) followed by visualization of the effect of the position of the colonies on the growth, tested in 96-colonies and 384-colonies formats. This step was conducted by first pinning the entire plate with one strain to characterize the position effect, including edge effect, followed by testing the growth of 91 MER strains. (3) Thereafter, we attempted to derive normalization tables, and (4) the efficacy of normalization to improve measurements was determined by measurement of four parameters: accuracy, precision, and the proportion of false positives and negatives by comparing them to previously reported spotting data [34].

**Figure 2 biomedicines-11-02829-f002:**
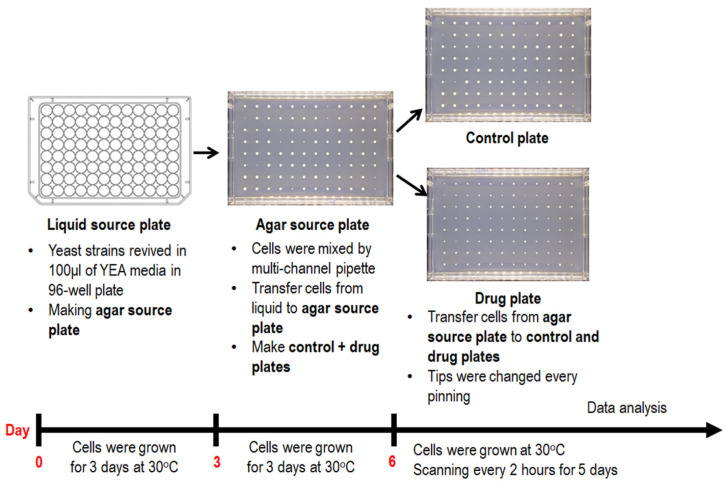
General procedure for ROTOR-based HU screening assay. Fission yeast cells of different genetic backgrounds revived in a YEA media in a 96-well plate were transferred onto a YEA agar source plate by Singer ROTOR and incubated at 30 °C for three days. Colonies grown on the source plate were later pin-transferred to control or drug plates, incubated at 30 °C and scanned with a Singer PhenoBooth^®^ machine every 2 h after the agar–agar transfer. Captured images were analysed with coupled PhenoSuite^®^ software (version 2.21.0304.1) to determine colony size for HU sensitivity evaluation.

**Figure 3 biomedicines-11-02829-f003:**
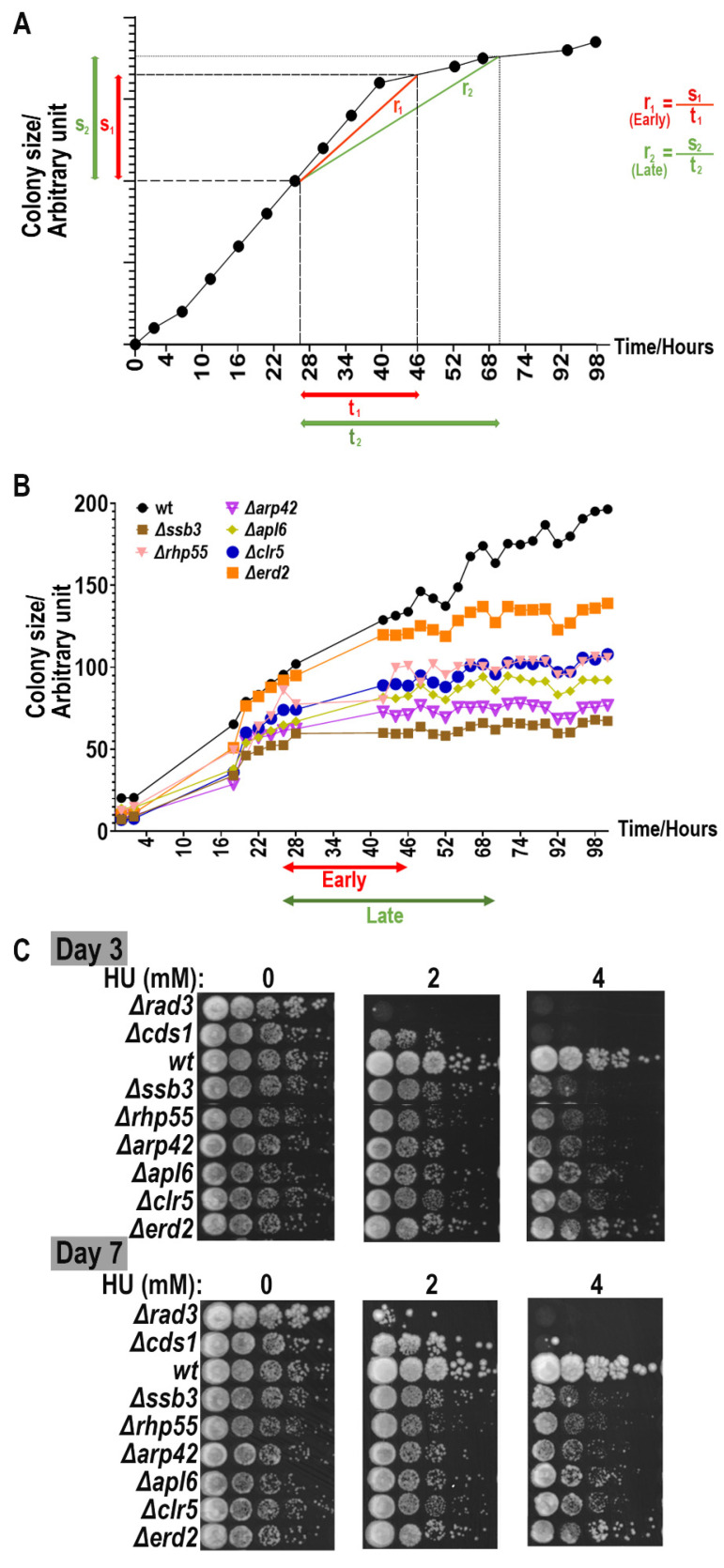
Analogizing traditional spotting analysis with ROTOR-based screening assay. (**A**) Growth curve (colony size vs. time) of WT fission yeast cells in ROTOR setting. Change in size of WT colony on ROTOR agar plate diminished with longer incubation hours (i.e., closer to lag phase), regardless of the presence or absence of HU. Colony size at a definite time point may not necessarily reflect the sensitivity of test strain against the candidate drug. Both relative growth rates of colonies at early log (r_1_) and late log growth phases (r_2_) are determined for better characterization of yeast cell sensitivity against HU. (**B**) Examples of growth curves generated with colony size values from a ROTOR-based screening with 4 mM of HU. Average colony size values of yeast strain determined by PhenoSuite^®^ software (version 2.21.0304.1) were plotted against incubation time after agar–agar transfer. Growth of fission yeast cells usually saturates at 55 h growth on agar plate in ROTOR setup, and hence average growth rates of strains of different genetic backgrounds were determined within the linear region of 5th to 55th h to characterize phenotypes of designated mutations. Strains that grew slower (e.g., *Δssb3* and *Δarp42*) were less likely to be differentiated at earlier time intervals after the strains were pinned onto the plates and consequently should be documented at the later time point. We further divided the linear range into early (27–46 h) and late growth (27–70 h) phases to better illustrate the differences in growth phenotypes. (**C**) Examples of spotting assay to determine drug sensitivity of deletion mutants (WT, *Δssb2*, *Δrhp55*, *Δarp42*, *Δapl6*, *Δclr5* and *Δerd2*) upon HU treatment (0, 2 and 4 mM). *Δrad3* and *Δcds1* strains were used as a positive control of the spotting. Cell growth is typically analyzed in spotting assay at two time points (i.e., days 3 and 7) after spotting onto agar media. Growth retardation for yeast cells on the drug plate at day 3 after spotting on the agar plate indicates that both yeast cell viability and growth are affected. Growth defects at day 7 after spotting on the agar plate indicate loss of cell viability in response to drug at testing concentration. Differences in early and late growth rates of deletion mutants from ROTOR screening generally followed the sensitivity pattern demonstrated in the spotting assay.

**Figure 4 biomedicines-11-02829-f004:**
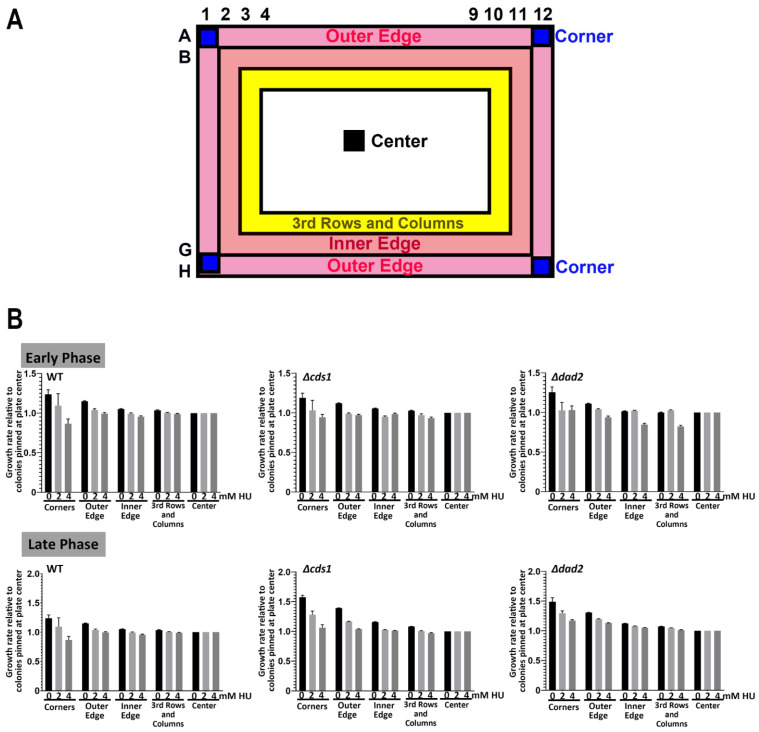
Minute “edge effects” observed at late phase in 96-spot ROTOR screening. (**A**) A map illustrating the relative position of “corners”,” outer edge”, “inner edge”, “center” and “3rd rows and columns” colonies at a 96-spot ROTOR screening setting. (**B**) WT, *Δcds1* and *Δdad2* cells did not show differences in relative growth rates at an early phase, regardless of pinning position and drug concentration. All 3 tested yeast strains showed a slight “edge effect” on the control plate at the late phase, with colonies grown on the four corners and the outer edge growing slightly better than cells at the center of the plate.

**Figure 5 biomedicines-11-02829-f005:**
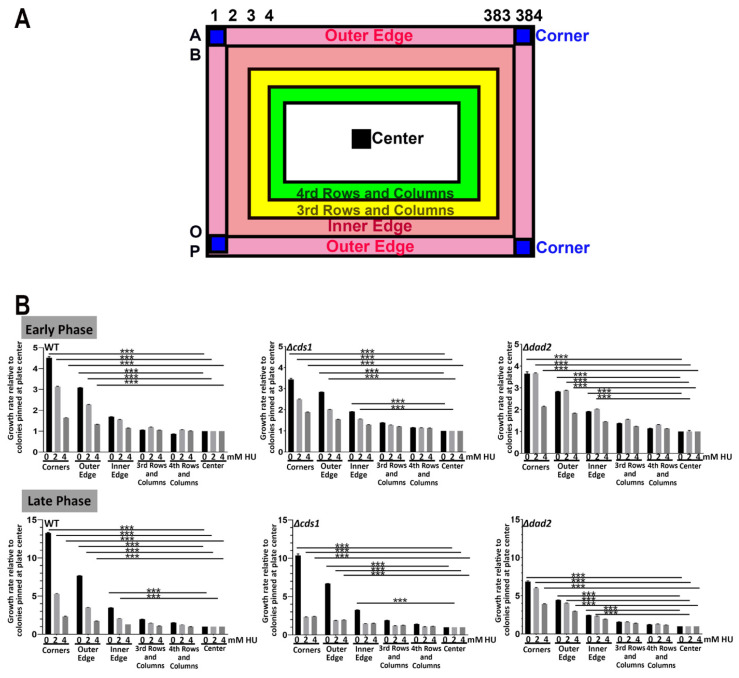
Extensive “edge effects” observed in 384-spot ROTOR screening, regardless of drug concentration. (**A**) A map illustrating the relative position of “corners”,” outer edge”, “inner edge”, “center”, “3rd rows and columns” and “4th rows and columns” colonies at a 384-spot ROTOR screening setting. (**B**) WT, *Δcds1* and *Δdad2* cells showed significant edge effects at both the early and late phases of growth and all drug concentrations. Fission yeast cells located at the corners and edges of agar plates gave a 4–13-fold higher relative growth rate than the same strains of cells pinned at the center of the YEA plate. *** *p* < 0.001, two-tailed *t*-test.

**Figure 6 biomedicines-11-02829-f006:**
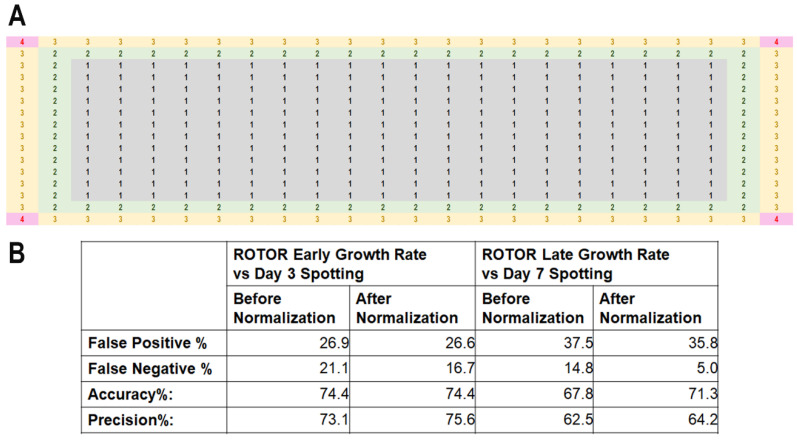
Growth rate adjustment with a normalization table improves the accuracy of ROTOR-based HU screening. (**A**) The normalization table was developed with reference to the relative growth rate of WT cells at different positions of the control plate. (**B**) Growth rate adjustment with the score table generally reduced the false-negative rate and improved accuracy of ROTOR screening of MER genes at the 384-spot setting.

## Data Availability

The manuscript contains all data pertaining to this work.

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
