# Peer review of "A Normalization Protocol Reduces Edge Effect in High-Throughput Analyses of Hydroxyurea Hypersensitivity in Fission Yeast"

_biomedicines, 2023, doi:10.3390/biomedicines11102829_

Round 1

Reviewer 1 Report

Draft titled"A normalization protocol reduces edge effect in high-throughput drug analyses in fission yeast" by Ulysses et al, Is a nice piece of work. The work may be important in solving practical problem while doing large scale screening. The overall draft is OK and I do not have any reservations against the manuscript as such. I have few suggestions and queries given below.

1) Why author choose S. pombe and not S. cerevisiae when S. cerevisiae is more commonly used for screening purpose. I think this need to be mentioned somewhere in draft.

2) To be sure that whatever results are obtained during screening, it is recommended that position of chemicals or strains under test should be changed to make sure that whatever is observed is not due to position specific. I think that should be discussed in draft at appropriate place.

3) Whether plates were incubated in humidified condition should be mentioned.

4) Reasons for choosing HU for experiment need to be discussed.

5) Generally when yeast-based screening are done, strains used are generally hypersensitive (deleted with PDR1, PDR3, PDR5). If these ORF are known or established in S. pombe???

6) Since plate reader based continuous growth measurement is more easy and fast, why author choose to measure size of colony. I think this need to be discussed.

Overall writing is pretty OK.

Author Response

Point-by-point response to REVIEWER 1’s comments

Comments and suggestions:
Draft titled "A normalization protocol reduces edge effect in high-throughput drug analyses in fission yeast" by Ulysses et al, Is a nice piece of work. The work may be important in solving practical problem while doing large scale screening. The overall draft is OK and I do not have any reservations against the manuscript as such. I have few suggestions and queries given below.

Response: Thank you very much for spending the time to review our manuscript. Thank you for the encouraging comments.

1) Why author choose  pombeand not S. cerevisiae when S. cerevisiae is more commonly used for screening purpose. I think this need to be mentioned somewhere in draft.

Response: We have mentioned several reasons for the benefits of doing drug hypersensitivity studies in fission yeast S. pombe in the introduction section: “Schizosaccharomyces pombe (S. pombe) is a powerful ascomycetes model organism possessing metazoan-like physiological processes, including human-like mitochondrial inheritance and energy metabolism, equal nuclear division through cell fission, regional centromere and other conserved genomic features. These processes are less conserved or compromised in budding yeast and other surrogate models used in screening. Fission yeast is also characterized by a high level of functionally homologous proteomic compositions and RNA interference pathway”. However, the main motivation for us to focus on S. pombe was that we possess highly reliable and reproduced data on HU hypersensitivity in S. pombe on strains that we have backcrossed to minimize the presence of other mutations, in particular to generate a prototrophic genetic background. The latter is for the purpose of removal of confounding false results, which we have reported would arise from nutritional marker mutations in an earlier paper (Tay et al, PLoS ONE, 2013. Doi: 10.1371/journal.pone.0055041)(Reference #58). Because our lab does not use S. cerevisiae, such optimized reagent and data set are unavailable for our standardization purposes. Thus S. cerevisiae was not employed in our analysis. We have included the following sentences into the end of the first paragraph of the Results section 3.1.1 to clarify this point: “Such spotting experiments performed in fission yeast have been robustly reproduced in our laboratory, especially on HU, thus establishing a reliable basis of comparison for the accuracy assessment on current high throughput pinning array analyses. Similar results performed on budding yeast were unavailable to us for verification purposes. As such, the pinning array analyses was not carried out with budding yeast.”

2) To be sure that whatever results are obtained during screening, it is recommended that position of chemicals or strains under test should be changed to make sure that whatever is observed is not due to position specific. I think that should be discussed in draft at appropriate place.

Response: Thank you for the comment. This is indeed an important considering. We have already performed the suggested experiment. Please refer to Figure S1-6, S8, S11, Figure 3-5. In these experiments, we have pinned the entire plates with one strain (instead of changing their positions on a plates pinned with many strains). The position effect has also been discussed in the Results as well as Discussion sections.

3) Whether plates were incubated in humidified condition should be mentioned.

Response: We did not incubate in a humidified condition. We have included this information into section 2.2.

4) Reasons for choosing HU for experiment need to be discussed.

Response: To address this comment, the sentences have been included in section 2.1: “HU was chosen for its water-solubility, chemical stability and heat resistance. Previously published hypersensitivity response of MER strains to HU [34] revealed by serial dilution spotting assay was used as the standard to compare the HU hypersensitivity response of the MER strains on the high throughput array.”

5) Generally when yeast-based screening are done, strains used are generally hypersensitive (deleted with PDR1, PDR3, PDR5). If these ORF are known or established in S. pombe???

Response: We are aware that many labs that do drug hypersensitivity screens do so by incorporating null mutations of genes of ATP-binding cassette drug transporter proteins such as budding yeast PDR1, PDR3 and PDR5 into the deletion mutant library. Similar treatment was performed for some studies in fission yeast as well, e.g. Arita et al (2011) Mol. Biosyst. Doi: 10.1039/c0mb00326c who incorporated null mutations of bfr1 (PDR5 homologue) and pmd1. However, we did not include these mutations into our strains. This is because in an earlier study of our lab, as mentioned above, we found even the supposedly ‘benign’ nutritional marker mutations were actually not so ‘benign’ as they confounded our screening outcomes by yielding more than a quarter false positive results (Tay et al (2013) PLoS ONE Doi: 10.1371/journal.pone.0055041). In the current study, determining the level of false positive and negative results is one of the most important aims and hence we deemed that a genetic background with as few unadulterated mutations as possible is essential for the accuracy of our study here.  

6) Since plate reader based continuous growth measurement is more easy and fast, why author choose to measure size of colony. I think this need to be discussed.

Response: We agree with the Reviewer that that plate reader based continuous growth measurement in liquid media does have its technical merits of speed and ease of use. However, the approach has significant drawbacks. This method has lower throughput than a high-density array based colony pinning approach. From our experience, growth data of suspension cultures (including that of unicellular fission yeast) can only be reliably obtained in 96 well format. Wells in higher density 384-well plate does not have sufficient circumference for the cultures to be thoroughly mixed, causing the cells to sink and collect as a layer at the bottom of the wells that greatly confound optical density measurement. Even for 96 well plates, very high speed shaking is needed (in the expense of risk of spilling and hence contamination of the cultures) to prevent the cells from sinking. Nevertheless, at high cell concentrations, even intense shaking becomes insufficient to prevent cell sinking in 96-well plates. Although these issues can be addressed by simply perform manually mixing of the cultures in the wells prior to plate reading (which we actually do also!), but doing so will be defeating the purpose of the high throughput approach with introduction of such laborious intervening manual step. We have included a section in the Discussion section 4.3 to discuss these points.

Reviewer 2 Report

In the reviewed manuscript, the authors describe in detail protocol for reduction of edge effect in high-through-put drug analyses in fission yeast. The manuscript is interesting from a technical point of view.

The authors should definitely shorten the manuscript and focus on the most important aspects.

The authors should prepare a clear flow chart (e.g., a graphical scheme). The developed protocol should be verified using several drugs. The image analysis protocol should be described in detail, including the background correction used.

The authors should verify their protocol on other yeast species as well as to answer how universal their protocol is for researchers using different yeast models. Performing such a comparison will also improve the novelty of the manuscript and increase the scientific importance of the present data.

Author Response

Point-by-point Response to REVIEWER 2’s Comments

Reviewer (2):

Comments and suggestions:
In the reviewed manuscript, the authors describe in detail protocol for reduction of edge effect in high-through-put drug analyses in fission yeast. The manuscript is interesting from a technical point of view.

Response: Thank you very much for the positive comment. Also thank you very much for spending the time to review our manuscript.

The authors should definitely shorten the manuscript and focus on the most important aspects.

Response: We have followed your suggestion to streamline the manuscript by removing the last part of the manuscript (corresponding to Figure S12) on assessing accuracy of the high-density growth assessment experiments using separate normalization tables at specific experimental conditions. Apart from this, we deem that the other portions of the manuscript are essential for our study. However, we do appreciate that the part on visualization of the position/edge effect, which constitute the major portion of the work, can be quite confusing, giving an impression that there were many apparently unrelated experiments. To address this, we thank you for your suggestion of incorporating a flow chart depicting the overview of our workflow, in your next comment. From the flow chart, it could be seen that there are 4 steps for the protocol development. The visualization of the position/edge effect was performed for 2 formats (96- and 384-format). Each format was done with either one strain on one plate, so as to quantify the edge effect; or different MER strains on one plate, to determine the efficacy of the normalization via assessment of the four parameters of precision, accuracy, % false positive or % false negative phenotypes. We hope that the incorporation of the flowchart can allow the relationship between the experiments to be more readily followed.

The authors should prepare a clear flow chart (e.g., a graphical scheme). The developed protocol should be verified using several drugs. The image analysis protocol should be described in detail, including the background correction used.

Response: Thank you again for your great suggestion of a flow chart. It has been incorporated as Figure 1. As for the screening of several drugs, we feel it is beyond our capability to do so in this work since our aim is to describe a workflow. We hope we would be able to do so in future work. As such we have modified the title of our paper to highlight the fact that the work has only been done for hydroxyurea. The modified title is “A normalization protocol reduces edge effect in high throughput drug analyses of hydroxyurea hypersensitivity in fission yeast”. We have included the detailed image analysis parameters and background correction in section 2.3.

The authors should verify their protocol on other yeast species as well as to answer how universal their protocol is for researchers using different yeast models. Performing such a comparison will also improve the novelty of the manuscript and increase the scientific importance of the present data.

Response: Thank you for this suggestion. However we view the suggested work too expansive beyond the capability of our current focus. The initial observation that has led us to the edge effect observation is based on the fact that we have gotten a very reliable set of low throughput spotting analyses for comparison with that obtained from the high-density study. These strains were extensively checked and backcrossed to minimize background mutations with special attention to constitute a prototrophic genetic background. Our spotting on particularly HU has been repeated extensively so we are very confident of how these mutants behave on HU. When the discrepancy arose on the high-density array, we were able to know immediately that that was a problem of the high-density method (as oppose to suspecting the authenticity of our spotting data). Thus to come to this extent of confidence and intricacy with other species is not possible with our current capability, time and resources availability. As such, we have stated in the title of the paper that this study is that of fission yeast. 

Round 2

Reviewer 2 Report

The authors addressed my comments and provided additional clarifications.